# TRACEBench: Personalized Function Calling Benchmark Based on Real-World Human Interaction

## ABSTRACT

Function calling has emerged as a central paradigm for augmenting the capabilities of Large Language Models (LLMs) by enabling them to transcend inherent limitations, particularly in accessing real-time information. Personalized tool utilization is essential for LLMs to adaptively select and invoke tools based on individual user profiles and historical interactions. However, most current benchmarks primarily rely on LLMs to simulate user interaction histories rather than using real-world interaction data, and these histories are typically short in length, offering limited evaluation of the model's ability to understand long-context inputs. In this work, we introduce TRACEBench, a benchmark designed to evaluate LLMs' function calling capabilities in terms of tool, parameter, and temporal context personalization. A significant difference from prior work is our data sourcing strategy: *TRACEBench is built upon authentic user interaction histories collected from human volunteers*, which provides a realistic foundation of user behavior and has been anonymized to protect user privacy. Furthermore, we build long tool-use records to facilitate the evaluation and optimization of tool-augmented models' long-context understanding capabilities. We perform reverse generation of user instructions from target tool calls, varying the level of instruction specificity to simulate different degrees of personalization. Extensive experiments offer insights into improving personalized LLM agents. Our code is available[1].

## 1 INTRODUCTION

Large Language Models (LLMs) have demonstrated remarkable capabilities in comprehending complex user queries and generating coherent responses (Zhao et al., 2023). However, their efficacy is constrained by fundamental limitations: their knowledge is static, and they often struggle with tasks demanding precise calculation or real-world interactions (Huang et al., 2025b; Chen et al., 2025). To surmount these challenges, function calling has emerged as a critical paradigm, enabling LLMs to access up-to-date information (Nakano et al., 2022), leverage specialized calculators (Gou et al., 2023; Das et al., 2024), and interact with thousands of real-world APIs (Mialon et al., 2023; Qin et al., 2023; Schick et al., 2023; Qu et al., 2025). Existing research has predominantly focused on improving what can be termed "general-purpose tool-use capability", which means the foundational mechanics of understanding tool functionalities and executing tasks based on explicit instructions (Xu et al., 2025). Yet, this focus overlooks a crucial aspect of practical application: personalization. In real-world scenarios, user intent is often conveyed implicitly and must be inferred from a user's unique profile, behavioral history, and latent preferences. This necessitates a shift in focus from general-purpose function calling to personalized function calling.

Comprehensively evaluating personalized function calling, however, presents a critical challenge. In this paper, we aim to construct a comprehensive benchmark for personalized function calling by holistically considering where personalization manifests. We identify three key dimensions:

- **Tool Personalization**, where users exhibit distinct preferences for functionally similar tools based on non-functional attributes like cost or quality of service. For instance, a user might prefer one food delivery app for its discounts but another for its speed, as illustrated in Figure 1(a).

---

[1] https://anonymous.4open.science/r/TRACEBench-5CC4

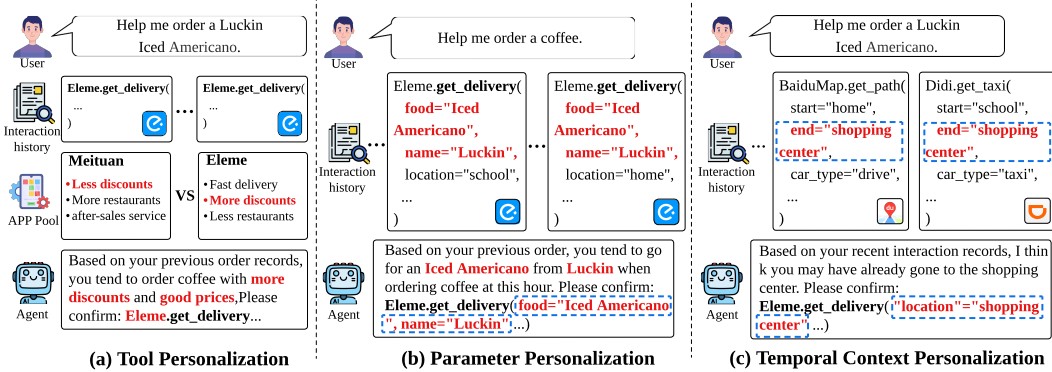

Figure 1: Three aspects of personalized function calling. (a) Tool Personalization: Users may prefer different tools for similar functionalities depending on the query context. (b) Parameter Personalization: Certain tool parameters may be missing from the user's query which need to be inferred from the historical usage of the similar tool (c) Temporal Context Personalization: Parameters needs to be inferred by understanding the spatio-temporal and logical relationships between different tools used over time.

- **Parameter Personalization**, which addresses the ambiguity in user expressions by inferring missing parameters from the historical usage of the similar tool and identify user's long-term, stable habits. In real-world scenarios, it's nearly impossible for users to express all their needs clearly in a single query. Instead, they often state their needs concisely, for example: "order me a coffee" vs "Order me a Luckin iced Americano for the office, with less sugar and light ice".

- **Temporal Context Personalization**, which requires reasoning over a user's short-term, dynamic context to infer parameters. This involves identifying logical and sequential dependencies between different, recently used tools. For example, a destination from a ride-sharing app should inform the location for a subsequent restaurant booking. This becomes especially complex within long (e.g., over 100 records) and noisy real-world interaction histories.

Existing benchmarks for personalized function calling typically focus on only one or two of these dimensions, as shown in Table 1. Current benchmarks (Xu et al., 2025; Huang et al., 2025b; Cheng et al., 2025) model personalization through static user profiles, neglecting the critical, dynamic nature of temporal context personalization. Furthermore, they often lack granular vagueness controls, unable to simulate user requests of different level of vagueness and are limited by short interaction histories. A more fundamental limitation lies in their data-sourcing methodology: Due to privacy and commercial concerns, real-world user interaction histories are not publicly available. Existing works rely on LLMs to simulate user profiles and interactions instead of real human interaction data. This discrepancy creates a significant gap between synthetic benchmarks and real-world applications. Such synthetic data struggles to capture the complexity of genuine human behavior and may suffer from preference homogenization (Schröder et al., 2025; Ding & Wang, 2025; Jiang et al., 2025), compromising the reliability of the evaluation.

To this end, we introduce the Tool, Parameter, and Temporal Context Personalization Evaluation Benchmark (**TRACEBench**), designed to facilitate a comprehensive and realistic evaluation of personalized function calling. A significant departure from prior work is our data sourcing strategy: **TRACEBench is built upon authentic user interaction histories collected from human volunteers**, which provide a realistic foundation of user behavior and has been anonymized to protect user privacy. To ensure the evaluation of tool preference, our benchmark is built upon diverse scenarios (e.g., e-commerce, transport) with multiple, functionally similar applications. For query generation, we employ a strategy of coarse screening followed by manual selection to identify anchor behaviors that highlight our three personalization dimensions. Finally, we use an LLM in a role-playing capacity to simulate user queries of varying vagueness, controlled by the number of explicitly provided parameters, enabling a granular assessment of LLM's ability to handle ambiguity.

The main contributions of this paper are as follows:

Table 1: Comparison of benchmarks across different criteria. The definition of personalization is given in section 3. "Long interaction History" refers to the benchmark provides a sufficiently long user interaction history—operationalized as at least 100. "Real Data" refers to the interaction history is sourced from real-world users and contexts, rather than synthesized or simulated.

| Function calling Benchmarks | Tool Personalization | Parameter Personalization | Temporal Context Personalization | Long Interaction History | Real Data |
|---|---|---|---|---|---|
| ToolBench (Qin et al., 2023) | ✗ | ✗ | ✗ | ✗ | ✗ |
| PeToolBench (Xu et al., 2025) | ✓ | ✗ | ✗ | ✗ | ✗ |
| ETAPP (Hao et al., 2025) | ✗ | ✓ | ✗ | ✓ | ✗ |
| PTBench (Huang et al., 2025b) | ✓ | ✓ | ✗ | ✓ | ✗ |
| **TRACEBench (Ours)** | ✓ | ✓ | ✓ | ✓ | ✓ |

- We provide a comprehensive discussion and formalization of personalized function calling, defining its core facets: tool, parameter, and temporal context personalization.

- We introduce *TRACEBench*, the first comprehensive benchmark for personalized function calling built on real human interaction data, designed to evaluate all three dimensions of personalization.

- We conduct extensive experiments on *TRACEBench*, providing a thorough analysis of various LLMs and offering valuable insights into the current challenges and future directions of personalized function calling.

## 2 RELATED WORKS

### 2.1 FUNCTION CALLING

The integration of large language models (LLMs) with external tools has emerged as a crucial capability for enhancing their capabilities in complex tasks. Existing approaches can be broadly classified into *tuning-free* and *tuning-based* methods. Tuning-free methods (Liu et al., 2024d; Shen et al., 2024; Yao et al., 2022) rely on in-context learning with prompt engineering techniques, such as chain-of-thought (Wei et al., 2022) prompting and demonstration ordering (Liu et al., 2024c), to guide LLMs in complex reasoning (Liu et al., 2024d; Shen et al., 2024). Tuning-based methods (Gao et al., 2024; Qin et al., 2023; Schick et al., 2024; Tang et al., 2023) directly finetune LLMs on specialized tool datasets, significantly enhancing their function-calling abilities through supervised training. Numerous benchmarks have been proposed to evaluate and improve the accuracy of function calling (Qin et al., 2023; Li et al., 2023; Wang et al., 2024). Existing benchmarks primarily evaluate general function calling performance. However, these benchmarks are designed for the general human population rather than for personalized function calling tailored to individual user preferences. Our work focuses on personalized function calling for LLMs.

### 2.2 LLM PERSONALIZATION

Personalization has become a critical research direction for LLMs, with active investigation across conversational AI (Cheng et al., 2024), recommendation systems (Huang et al., 2025a; He et al., 2023), information retrieval (Zhou et al., 2024), and education (Kasneci et al., 2023; Park et al., 2024). This focus is now extending to the domain of function calling. Existing works have begun developing benchmarks for personalized function calling (Hao et al., 2025; Xu et al., 2025; Cheng et al., 2025; Huang et al., 2025b). For instance, ETAPP (Hao et al., 2025) partitions an agent's memory into long-term user preferences and short-term states. PeToolLLM (Xu et al., 2025) classifies personalization into implicit user preferences and non-functional tool attributes. ToolSpectrum (Cheng et al., 2025) identifies user profiles and environmental conditions as key personalization factors for generating user queries. PTBench (Huang et al., 2025b) defines personalization through user tool preferences and profile-dependent queries. However, these benchmarks overlook **temporal context personalization**, the ability to reason about the logical and sequential dependencies between different, recently used tools. Furthermore, they rely on LLMs to simulate user profiles and interaction data, and they lack a fine-grained, quantitative classification of query vagueness.

TRACEBench is designed to address these gaps. By introducing temporal context personalization dimension, grounding benchmark in real human interaction histories, and generating queries with

different level of vagueness. We propose the first comprehensive benchmark for personalized function calling built on real human interaction data, designed to evaluate all facets of personalization.

## 3 PRELIMINARIES

In this section, we formalize the task of function calling, which requires the model to leverage user-specific information when selecting and configuring tools to address user needs. We first define *general-purpose function calling*, which represents the standard paradigm in existing research. We then build upon this foundation to introduce and formalize *personalized function calling*, systematically defining it along the three key dimensions established in our introduction: 1) Tool Personalization, 2) Parameter Personalization, and 3) Temporal Context Personalization.

### 3.1 GENERAL-PURPOSE FUNCTION CALLING

Standard function calling aims to select an appropriate tool and its corresponding parameters from a set of candidates to fulfill a user's request. Formally, given a user query $q_u$ and a candidate tool set $\mathcal{T} = \{d(t_1), d(t_2), \ldots, d(t_N)\}$, where $d(t_i)$ is the documentation for tool $t_i$, a LLM with parameters $\theta$ is tasked with generating a tool call $c = (t, p)$, where $t \in \mathcal{T}$ and $p$ is the set of its corresponding parameters. And $P$ is the likelihood of generating the tool call $(t, p)$. This can be modeled as:

$$\hat{c} = (\hat{t}, \hat{p}) = \arg\max_{(t,p)} P((t, p)|q_u, \mathcal{T}; \theta) \tag{1}$$

Personalized function calling extends this formulation by conditioning the generation on user-specific information. We introduce a dynamic interaction history $\mathcal{H}_u$, which is an ordered sequence of past tool calls and their outcomes. The personalized function calling task is then to generate a tool call $(t_u, p_u)$ that is not only functionally correct but also optimally aligned with the user's implicit and explicit preferences:

$$(\hat{t}_u, \hat{p}_u) = \arg\max_{(t,p)} P((t, p)|q_u, \mathcal{T}, \mathcal{H}_u; \theta) \tag{2}$$

### 3.2 TOOL PERSONALIZATION

This personalization focuses on scenarios where multiple tools, often from different platforms, possess equivalent core functionalities but differ in their non-functional attributes (e.g., price, delivery speed, maintenance service). The model must resolve this ambiguity by selecting the tool that best aligns with the user's context-dependent preferences. For example, a user may prefer a platform with superior service for purchasing high-value electronics but prioritize a cheaper, faster platform for daily necessities (Huang et al., 2025b). We term this phenomenon as tool personalization.

**Definition 3.1 (Tool Personalization).** For a user $u$, query $q$, and a set of functionally equivalent tools $\{t^1, t^2, \ldots\} \subset \mathcal{T}$, a preference relation $\succ_{(u,q)}$ exists such that $t^i \succ_{(u,q)} t^j$ indicates that tool $t^i$ is preferred over $t^j$ for this specific context. The objective is to select a tool $\hat{t}$ that is maximal with respect to this preference relation, i.e., $\neg\exists t' \in \mathcal{T}$ such that $t' \succ_{(u,q)} \hat{t}$.

### 3.3 PARAMETER PERSONALIZATION

In real-world interactions, user queries are often concise and omit necessary details, leading to incomplete queries. Parameter personalization involves inferring missing parameter values by identifying a user's **long-term preferences** from their interaction history. This form of reasoning typically relies on patterns established over many interactions with **same or functionally similar tools**.

**Definition 3.2 (Parameter Personalization).** Given a query $q_u$, a user interaction history $\mathcal{H}_u$, and the ground-truth parameter values $A$ for the solution, the query is history-dependent if there exists a parameter value $\alpha \in A$ such that its informational content is not present in the query but is derivable from the interaction history, i.e., $\alpha \notin \text{span}(q_u) \wedge \alpha \in \text{span}(\mathcal{H}_u)$, where $\text{span}(\cdot)$ denotes the set of information conveyed.

Figure 2: The overall framework of TRACEBench.

## 3.4 TEMPORAL CONTEXT PERSONALIZATION

In contrast, temporal context personalization involves inferring parameters from **short-term, dynamic context**. It requires the model to identify logical and sequential dependencies between **different, recently used tools**, capturing evolving, multi-step user intents. For example, the destination from a recently used ride-sharing app informs the location for a subsequent restaurant booking.

**Definition 3.3 (Temporal Context Personalization).** Let the interaction history be an ordered sequence arranged in chronological order $\mathcal{H}_u = \langle (c_1, o_1), \ldots, (c_{r-1}, o_{r-1}) \rangle$, where each tool call is $c_i = (t_i, p_i)$. The current tool call $c_r = (t_r, p_r)$ exhibits temporal context personalization if a parameter value $\alpha \in p_r$ is the parameters of a function from the recent history or can be inferred from the recent history through logical reasoning, and it does not belong to the user's long-term profile parameters., $\alpha = f(\{p_{r-k}, \ldots, p_{r-1}\})$, where $\alpha$ is neither specified in the current query $q_r$ nor present in the static user long-term interests.

## 4 TRACEBENCH

To address the three personalization challenges in personalized function calling mentioned above, we introduce TRACEBench, a benchmark and data generation framework designed for generating high-quality training and evaluation data for personalized function calling. This section details our three-stage construction pipeline: 1) Hierachical Function Preparation, 2) Real-World Data Collection, and 3) anchor behavior selection and query generation, as illustrated in Figure 2.

### 4.1 TOOL PREPARATION

A diverse and realistic toolset is fundamental to evaluating personalization. We identified 8 common application scenarios based on prior work (Cheng et al., 2025; Huang et al., 2025b) and app analysis, including shopping, food delivery, reminder, transport entertainment and so on. For each domain, we collected and adapted APIs from multiple, distinct real-world applications. This diversity is crucial for evaluating **Tool Personalization**, where an agent must choose between functionally similar tools based on latent user preferences. For example, in the shopping domain, users can choose between JD or Taobao at the APP level to meet the same shopping needs. In each scenario, we leverage GPT-4o to generate initial designs and tool descriptions for Apps and APIs with similar functions. Then we manually check and optimize the output to ensure they accurately reflect real-world functionalities. All collected APIs were standardized, ensuring consistent documentation structure, parameter naming conventions, and functionality descriptions. All Apps and APIs are listed in Table 5.

### 4.2 DATA COLLECTION AND PREPROCESSING

Personalization requires diverse and realistic user history. The authenticity of user interaction data is the cornerstone of our benchmark. Existing benchmarks for personalized tool learning, such as PTBench Huang et al. (2025b) and PEToolBench Xu et al. (2025), primarily rely on LLMs to

synthesize both user profiles and their corresponding interaction histories. While this approach circumvents significant privacy and data collection challenges, it introduces critical limitations. We argue that LLM-synthesized data lacks the necessary realism for a stringent evaluation. In real-world applications, an agent does not have access to a structured user history API due to commercial and privacy constraints; it can typically only observe a user's instruction history. LLM-generated histories serve as a poor proxy for this reality, as they often exhibit repetition, lack long-term logical consistency, and fail to capture the subtle, non-linear evolution of genuine human behavior.

To address these shortcomings, our benchmark is built upon real, anonymized interaction data collected over several weeks from a cohort of human volunteers. This approach ensures that the underlying data reflect authentic user preferences, habits, and temporal dynamics. The collected data underwent a rigorous preprocessing pipeline:

- **Anonymization:** All personally identifiable information (such as name, age, etc.) was scrubbed from the logs to ensure the privacy of participant.
- **Validation:** The interaction logs were validated for temporal and logical consistency. Incomplete or corrupted sessions were discarded to maintain the integrity of the dataset.
- **Standardization:** All logged interactions were mapped to the standardized API formats defined during tool preparation, creating a clean, structured dataset of real user-tool interactions over time.

### 4.3 Anchor Behavior Selection & Query Generation

**Anchor Behavior Selection.** Given a long and complex interaction history for each user, a critical challenge is to identify which behaviors represent meaningful tests of personalization. We therefore developed a two-stage methodology to select high-value anchor behavior. The first stage involves **coarse-grained filtering** based on predefined policies designed to detect unusual or historically relevant behaviors in a user's behavior. We suppose that the most interesting personalization challenges arise from short-term deviations from a user's established patterns or behaviours that logically related to recent interactions. Our filtering policy identifies anomalous parameter values within the user's timeline or parameters that are logically related to recent history. For example, a user previously taks a taxi to a certain location and then searched for a restaurant there, where the location differs significantly from their usual places. In the second stage, **fine-grained selection**, these candidates are first screened by an LLM and then manually reviewed by human annotators to verify their quality and ensure they represent temporal context personalization challenges.

**Query Generation.** For each validated anchor behavior, we generated corresponding natural language queries. For this task, we leveraged the role-playing capabilities of advanced LLMs, a common practice in benchmark creation (Huang et al., 2025b; Wang et al., 2025; Cheng et al., 2025). The LLM is prompted with the anchor behavior selected in previous step, along with the user's interaction history leading up to that point. A core innovation of our framework is the introduction of a **quantifiable vagueness levels** for the generated queries, a feature absent in prior benchmarks. We define query vagueness as a function of the number of parameters explicitly specified in the query text, where a smaller number of explicitly included parameters indicates a higher level of vagueness. The LLM was explicitly instructed to generate queries at four distinct vagueness levels:

- **Most Clear:** The query contains a complete and unambiguous set of parameters for executing the intended function calling, excluding any elements that may introduce ambiguity.
- **Relevant Clear** The query explicitly includes most of the parameters required to execute the target function call. In this paper, we define this as omitting only one or two parameters (including the function name itself) relative to the ground-truth call.
- **Relevant Vague:** The query omits some key parameters, requiring the model to infer them from a recent context or stable, long-term preferences. In this paper, we define this as omitting half of the parameters relative to the ground-truth call.
- **Most Vague:** The query is highly implicit, containing no explicit parameter specifications. The model must infer them by integrating evidence from the user's interaction history and conducting logical reasoning.

Finally, to ensure evaluation accuracy, all generated queries, along with their corresponding user histories and ground-truth function calls, are subjected to a final stage of manual review and annota-

tion. This process verified the naturalness of the queries, the correctness of the ground-truth labels, and the appropriateness of the assigned vagueness level.

## 4.4 DATASET DETAILS

We leverage GPT-4o to synthesize the personalized function calling dataset via our framework. The overall dataset consists of a total of 10 users and 1815 queries under 8 scenarios, including shopping, food, reminder, transport, news, knowledge, entertainment and sport, featuring 38 APPs, 55 functions, a total of 312 parameters, and a total of 94 required parameters. To ensure the quality of the test set, we manually verify each sample. The statistics are illustrated in Table 2. In addition, we provided a detailed display of the number of apps, functions, average parameters, and required parameters in each of scenarios. The statistics are illustrated in Table 3.

Table 2: Statistics of the datasets.

| Dataset | #Scenario | #APPs(Tools) | #Functions | #User | #Query |
|---|---|---|---|---|---|
| TRACEBench | 8 | 38 | 55 | 10 | 1815 |

Table 3: Statistics of the datasets (by Scenario).

| Scenario | #APPs(Tools) | #Functions | avg #param | avg #required |
|---|---|---|---|---|
| Food | 7 | 11 | 9 | 2 |
| Shopping | 3 | 3 | 6 | 1 |
| Transport | 12 | 13 | 6.38 | 3.23 |
| Reminder | 1 | 3 | 3 | 2.67 |
| Entertainment | 8 | 15 | 5.13 | 0.67 |
| Sport & Health | 2 | 4 | 2.75 | 1.5 |
| Knowledge | 3 | 3 | 4 | 1 |
| News | 3 | 3 | 1 | 0 |

## 5 EXPERIMENT

### 5.1 EXPERIMENTAL SETTINGS

**Evaluation Metrics.** This paper follows prior research (Huang et al., 2025b). We first check format accuracy to determine if the output accords with the required template, reflecting instruction-following ability. Each output must include app name, function name, and function calling parameters. We compute accuracy separately for the function name, parameters names, and parameters values, and then report an overall accuracy. The calculations of the metrics is shown in Table 5.

**Models.** We compare the latest open-source models and API-based models. API-based models include GPT-4o, Deepseek-v3 (Liu et al., 2024a), Deepseek-r1 (Guo et al., 2025). Models fine-tuned for function-calling include Hammer2.1-7b (Lin et al., 2024), ToolACE-2-Llama-3.1-8B (Liu et al., 2024b), watt-tool8B, Arch-Agent-7 and xLAM-7b-r (Zhang et al., 2024). Open-source models include Qwen2.5-7B-Instruct (Team, 2024), Llama-3.1-8B-Instruct and Mistral-7B-Instruct-v0.3 (Lyu et al., 2023).

**Implementation Details.** For all models, we set the temperature to 0.1 and top-p to 0.1 to minimize stochastic variations in the output, ensuring a consistent evaluation of model performance. Open-source models are evaluated on NVIDIA 4090 GPUs, while API-based models are assessed through direct API calls to OpenAI.

### 5.2 MAIN RESULTS

We evaluate the performance of different models on TRACEBench and report the results in Table 4. We have the following findings according to the results:

**General Conclusion on Model Performance.** API-based closed-source models significantly outperform smaller open-source models across various dimensions, including format compliance, tool personalization, parameter personalization abilities. This aligns with the conclusions from most benchmarks, a result mainly attributable to the enhanced capabilities stemming from the model's larger scale of parameters. However, the performance gap between closed-source models and leading open-source models such as Qwen2.5-7B-Instruct is progressively diminishing. This convergence is largely propelled by advancements in model architecture and training methodologies, signaling that open-source solutions are steadily approaching parity with closed-source models.

**Degradation of Generalization in Fine-Tuned Models.** Within the open-source LLMs, general-purpose instruction-tuned models such as Llama-3.1-8B-Instruct, demonstrated relatively good per-

Table 4: Evaluation results of different open-source and closed-source models on TRACEBench. Bold and underline represent the best and the 2nd best results. Function_param_names denotes the ability of output correct parameter name in function calling.

| Type | Model | Format | APPs(Tools) | function_names | function_param_names | function_param_values | overall |
|---|---|---|---|---|---|---|---|
| API | **GPT-4o(FC)** | 1.0000 | 0.8512 | 0.9295 | 0.6121 | 0.3548 | 0.3267 |
| | **Deepseek-r1(FC)** | 0.9912 | 0.8628 | 0.9140 | 0.4187 | 0.2402 | 0.2292 |
| | **Deepseek-v3(FC)** | 0.9477 | 0.8055 | 0.8777 | 0.3857 | 0.2342 | 0.2220 |
| Open-Source | **Qwen2.5-7B-Instruct** | 1.0000 | 0.7802 | 0.9129 | 0.4121 | 0.2193 | 0.2017 |
| | **Llama-3.1-8B-Instruct** | 0.9879 | 0.7778 | 0.8953 | 0.5008 | 0.2623 | 0.2479 |
| | **Mistral-7B-Instruct-v0.3** | 0.5699 | 0.4380 | 0.5163 | 0.2204 | 0.1113 | 0.0986 |
| | **Hammer2.1-7b** | 0.9879 | 0.6639 | 0.8970 | 0.3708 | 0.1983 | 0.1664 |
| | **ToolACE-2-Llama-3.1-8B** | 0.5366 | 0.3796 | 0.4733 | 0.1730 | 0.0793 | 0.0716 |
| | **Watt-tool-8B** | 0.5229 | 0.3879 | 0.4645 | 0.2187 | 0.1157 | 0.1085 |
| | **xLAM-7b-r** | 0.5063 | 0.3763 | 0.4358 | 0.2039 | 0.1146 | 0.1063 |
| | **Arch-Agent-7B** | 0.3895 | 0.2595 | 0.3631 | 0.1636 | 0.0744 | 0.0595 |

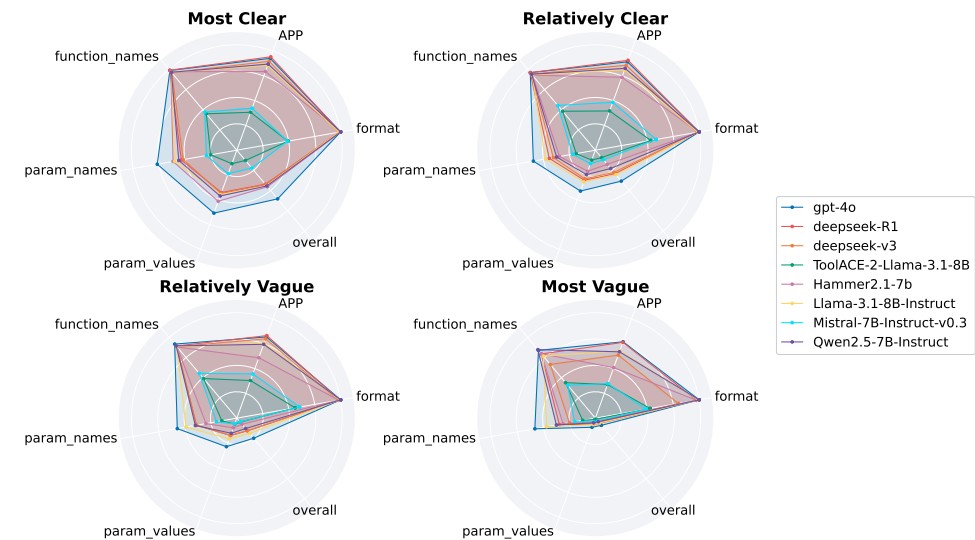

Figure 3: The performance of open-source and close-source LLM on different testing subsets w.r.t. the query difficulty per query on TRACEBench.

formance, achieving overall accuracies of 0.2418. In contrast, models fine-tuned on function-calling datasets, like Hammer2.1-7b, exhibited comparatively weaker performance. This may indicate a phenomenon of "generalization loss". Specifically, while fine-tuning on specialized datasets can enhance performance on targeted tasks, it may degrade the model's ability to generalize to broader instructions or functions from different domains.

**Performance Limitations in Parameter Personalization.** Across all models, we can observe that the performance drops sharply when the task involves populating parameter values. While top models can select correct APP and function name with high accuracy, performance sharply drops on predicting parameter values. This result indicates that generating personalized parameters is a more difficult task for most models. In essence, the core difficulty in function calling lies in understanding user intent and precisely extracting or inferring parameters from user interaction history.

### 5.3 ANALYSIS OF QUERY VAGUENESS

To further investigate how the model performs under varying levels of personalization requirements, we classify TRACEBench by the four levels of query vagueness defined in Section 4.3 and conduct experiments. As shown in Figure 3, a clear trend of performance degradation is observed across all models as the query vagueness increases. This visually demonstrates that the task becomes substantially more challenging when the model is required to infer more implicit information from user history and context, leading to a general decline in the performance of existing models.

Besides, the performance drop is most significant for **function_param_values**, which drops rapidly as queries become more vague. This observation, consistent with our main findings in Table 4, highlights that the core challenge in personalized function calling lies in the deep semantic understanding and inference of user intent. We define "parameter personalization" and "temporal context person-

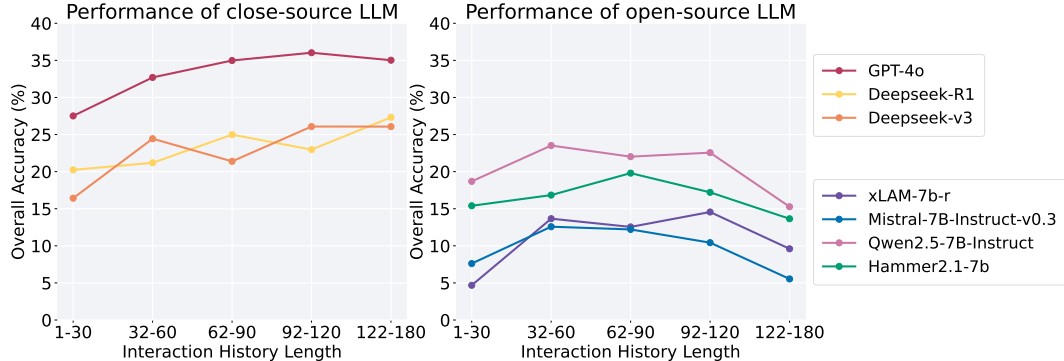

Figure 4: The performance of open-source and close-source LLM on different testing subsets w.r.t. the length of interaction history per query on TRACEBench.

alization" as two core components of personalized function calling. High-difficulty queries are inherently highly implicit, making them suitable for testing these two capabilities. The experimental results show a steep performance drop in current models when faced with such queries, directly reflecting their significant deficiencies in effectively leveraging users' long-term preferences and short-term dynamic contexts to infer missing parameters and lack the robust reasoning capabilities.

### 5.4 IN-DEPTH ANALYSIS

A key feature of TRACEBench is its long user interaction histories. To analyze the impact, we partition the test set into subsets based on interaction history length of each user query and evaluate LLMs on these subsets. As shown in Figure 4. We can observe that for the vast majority of models, including GPT-4o in closed-source LLMs and most open-source LLMs, performance initially rises as the interaction history grows. However, after reaching an optimal length, the performance deteriorates as the history record length continues to increase. Besides, for reasoning models like Deepseek-r1, overall performance does not show a significant decrease as the history length increases.

This is because, initially, as contextual data increases, additional histories helps to infer latent tool preferences, providing more evidence for personalized function calling decisions and enabling the model to better understand user's specific intent. However, as the interaction history length extends further, models hit the wall in effectively processing and utilizing long interaction histories. They may struggle to distinguish relevant past interactions and be affected by the user's use of massive different functions, thereby weakening their ability to accurately understand the user's current intent. This reveals a potential limitation in the models' long-context understanding and reasoning capabilities. Once the cognitive limit for processing the history is surpassed, the model's performance in personalized function calling begins to deteriorate. For reasoning models, its reasoning capabilities meet the requirements of temporal context personalization, enabling it to better understand user intent and perform temporal logic reasoning over long interaction histories.

## 6 CONCLUSION

In this work, we introduce the concept of personalized function calling and proposed three personalization challenges: tool personalization, parameter personalization and temporal context personalization. These tasks require the model's ability to select preferred tools, inferring parameter values from long-term habits and reasoning over recent short-term user interact history. To evaluate LLMs' personalized function calling capabilities, we introduced TRACEBench, a new benchmark grounded in real-world human interaction data. Extensive experimental evaluations assess the personalized function calling abilities of existing models, confirming the effectiveness of our synthesized data. Experiments reveal that current LLMs struggle significantly with inferring correct parameter values. This highlights that the core challenge lies in the deep understanding of user intent and temporal logical reasoning. These findings guide future work toward enhancing models' deep reasoning and long interaction history abilities, which will help to enhance the personalization capabilities of LLMs.

## REPRODUCIBILITY STATEMENT

To ensure the reproducibility of our work, we have made our experimental setup and data construction methodology fully transparent. We provide detailed experimental setup, including model details and hyperparameters, in Section 5.1. For the convenience of review, all code and data are available at the following anonymous repository: `https://anonymous.4open.science/r/TRACEBench-5CC4/`, and it will be made publicly available in the future. The repository includes a detailed README file, which contains clear instructions for reproducing our results.

## REPRODUCIBILITY STATEMENT

The development of TRACEBench was guided by a strong commitment to ethical research practices. The benchmark is built upon authentic user interaction histories collected from a cohort of human volunteers. We assure our volunteers that their data will be rigorously anonymized and will not be used for any commercial purposes. To protect the privacy of these participants, all data underwent a rigorous anonymization process where personally identifiable information (e.g., name, age) was scrubbed from the interaction logs. While our code and data will be made publicly available to encourage reproducibility, the raw, anonymized, user interaction logs will not be released to protect the privacy of our volunteers. We acknowledge that our volunteer cohort may not be representative of the global population, and the dataset may contain inherent demographic or behavioral biases; future work should aim to broaden participant diversity. The goal of this research is to evalute the capabilities of personalized LLM agents.

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

# A  THE USE OF LLM

In this paper, Large language models (LLMs) were used as a general-purpose assist tool to improve the clarity and grammar of the manuscript. The models were not used for research ideation, data analysis, or the generation of any core content. Their role was limited to minor editing and polishing of the text to enhance readability.

# B  DATASET DETAILS

This section shows all Apps and APIs defined in our benchmark.

Table 5: List of all Apps and their corresponding functions in the TRACEBENCH.

| Domain | APPs | APIs |
|---|---|---|
| Shopping | Temu, Amazon, Poizon, Vipshop, Xianyu | *getProductList*, *buyProduct* |
| Travel | Baidu_Maps, Didi_Chuxing, Ctrip | *getDistance*, *getRoute*, *bookTaxi*, *rentCar*, *bookTicket*, *bookHotel* |
| Entertainment | Maoyan, Damai | *getShowSchedule*, *bookShowTicket* |
| Grocery | Freshippo, Duoduo Maicai | *getProductList*, *buyProduct* |
| Delivery | Cainiao Guoguo | *createShipment*, *getShipmentStatus*, *getCourierLocations* |
| Finance | Bank, Tonghuashun | *getFundDetails*, *buyFund*, *getStockDetails*, *buyStock* |
| Health | Ping An Health, Keep | *createHealthPlan*, *logExercise* |
| Knowledge | Xiaohongshu, Zhihu, Dedao | *getKnowledge* |
| News | Toutiao, Weibo, Hupu | *getDailyNewsRecommendations* |

# C  EXPERIMENT SETUP DETAILS

## C.1  EVALUATION METRICS

The calculation of various metrics in PTBench are formulated as follows:

- **Format Accuracy** refers to the proportion of the LLM's generated output in conforming to our required output template, which indicates the instruction-following ability.

$$\text{format\_acc} = \frac{\#\text{parsable samples}}{\#\text{total samples}} \tag{3}$$

- **APP Accuracy** refers to the proportion of function calls generated by the LLM where the selected app is the same as the ground-truth tool, which indicates the tool comprehension ability.

$$\text{APP\_acc} = \frac{\#\text{correct APP samples}}{\#\text{total samples}} \tag{4}$$

- **Function Name Accuracy** is the proportion of samples where the function name in an LLM-generated tool call is an exact match to the ground truth, which indicates the function selection ability.

$$\text{function\_name\_acc} = \frac{\#\text{correct function name samples}}{\#\text{total samples}} \tag{5}$$

- **Function Parameter Accuracy** is the proportion of samples where both the number and the names of the parameters in an LLM-generated tool call are an exact match to the ground truth.

$$\text{function\_param\_acc} = \frac{\#\text{correct function param samples}}{\#\text{total samples}} \tag{6}$$

- **Function Parameter-Value Accuracy** is the proportion of samples where the parameter names and their corresponding values in an LLM-generated tool call are both an exact

match to the ground truth, which indicate the parameter personalization and temporal context personalization ability.

$$\text{function\_value\_acc} = \frac{\#\text{correct parameter value samples}}{\#\text{total samples}} \tag{7}$$

- **Overall Accuracy** indicate the overall personalized function calling ability.

$$\text{overall\_acc} = \frac{\#\text{full correct samples}}{\#\text{total samples}} \tag{8}$$