# OpenReview forum: "TRACEBench: Personalized Function Calling Benchmark Based on Real-World Human Interaction"
_ICLR.cc/2026/Conference — ICLR 2026 Conference Withdrawn Submission_

### Official Review · Reviewer_ygdZ · 2025-10-26

**Soundness:** 2
**Presentation:** 2
**Contribution:** 1
**Rating:** 2
**Confidence:** 4

**Summary:**

- This paper introduces TRACEBench to evaluate LLMs function / tool calling abilities in personalized and long-context scenarios. It notes the limitation of existing benchmarks that relying on short, simulated user histories by instead using real, anonymized interaction data from human volunteers.
- Introduced benchmark assesses personalization across tool selection, parameter use, and temporal context, and includes reverse generated user instructions with varying specificity to test how well models adapt to different personalization levels.
- The benchmark is across 3 dim viz. Tool Personalization (implicit user preference from historical data), Parameter Personalization (lack of info or details / under constructed user queries) and Temporal Context Personolization (dynamic preference based on short term history, like recently used tools)

**Strengths:**

- Paper actually uses real user data from human users so this should reflect in the quality of their benchmark against existing benchmarks.
-

**Weaknesses:**

See questions

**Questions:**

1. My main qualms with this work is the motivation. Tool calling and user personalization of LLM's responses are two orthogonal items. They should be evaluated on them differently. I don't find this benchmark useful and well motivated. In the example 1, in Fig 1, I find it unrealistic that user provides instruction to order an "Luckin Iced Americano" but w/o mentioning via which app (so tool can be selected). Like, then why not also infer which store for "Iced Americano" in this case. I'm sorry, but this is not making sense to me. If the example was, where user themselves have little idea on their preference and not just "its implicitly buried" then maybe I would understand. But tools are general things, and they're only handfuls that are overloaded for many users. So this doesn't make sense to me.
1. How is Temporal Context Personalization (TCP) is not subset of Tool Personalization (TP)? Aren't they both based on user pref and their past data, just that TCP looks only at most recent things and TP at everything from past / user info.
1. In Table 4, nothing is underlined or bold, even though the caption says they mean best and 2nd best respectively.
1. The results miss many models from both closed and open source like GPT5, Gemini, Claude, Qwen3, etc.

**Details Of Ethics Concerns:**

This paper contains data from actual human volunteers. I am marking this to let S/AC know since I'm not aware how it works. This does not effect my review score or points whatsoever.

---

### Official Review · Reviewer_zaJH · 2025-10-31

**Soundness:** 3
**Presentation:** 3
**Contribution:** 3
**Rating:** 6
**Confidence:** 4

**Summary:**

This paper focuses on the problem of personalization in LLM function calling, arguing that current research predominantly focuses on general-purpose tool use while overlooking user-specific needs. It is the first to systematically propose and formalize three types of personalization: Tool Personalization, Parameter Personalization, and Temporal Context Personalization. The authors construct the TRACEBench benchmark based on real, anonymized user interaction histories. A multi-stage pipeline (tool preparation, data collection, anchor behavior selection, and query generation) is used to create the dataset. Extensive experiments evaluate numerous open-source and closed-source models, revealing significant shortcomings in current models, especially in inferring parameters and leveraging long interaction histories.

**Strengths:**

1.The paper provides the first clear and structured definition of personalized function calling, categorizing it into three distinct dimensions (Tool, Parameter, Temporal Context) with formal definitions. This establishes a solid theoretical foundation for future research in this area.
2.TRACEBench is built upon genuine, anonymized user interaction logs, unlike many existing benchmarks that rely on LLM-simulated data. This significantly enhances the benchmark's reliability, realism, and practical relevance.
3.The experimental design is thorough. Beyond overall performance, it includes valuable analyses of the impact of query vagueness and interaction history length on model performance. These analyses effectively pinpoint the current bottlenecks in models for personalized function calling.

**Weaknesses:**

1.While the use of real data is commendable, the dataset is relatively small, comprising only 10 users and 1,815 queries. This limited scale may affect the generalizability of the evaluation results and their statistical significance.
2.Although the introduction of quantifiable vagueness levels is innovative, the queries themselves are synthesized by an LLM. This approach might introduce model-specific biases, potentially compromising the objectivity of the evaluation.
3.While Temporal Context Personalization is proposed as a core dimension, the experiments do not include targeted comparisons or a separate, detailed analysis to specifically validate and dissect model performance on this capability. The discussion of results for this aspect remains somewhat high-level.

**Questions:**

1.	The dataset currently includes only 10 users. Are there plans to expand the user base to improve diversity and representativeness? Is the inclusion of cross-cultural or multilingual user data being considered for future work?
2.	During the query generation process, how did you ensure semantic and logical consistency across different vagueness levels? Was human validation or any other method employed to control and verify the quality of the generated queries?

---

### Official Review · Reviewer_PWPU · 2025-10-31

**Soundness:** 2
**Presentation:** 2
**Contribution:** 2
**Rating:** 0
**Confidence:** 5

**Summary:**

The main contents and contributions of this paper can be summarized as,

Function calling lets LLMs use tools in real time, but most benchmarks rely on short, simulated histories, limiting evaluation of personalization and long-context understanding. This paper introduces TRACEBench, built from anonymized, authentic interaction histories and long tool-use records, to assess tool, parameter, and temporal-context personalization. This paper reverse-generates instructions from target tool calls at varying specificity and reports experiments that reveal practical insights for designing personalized LLM agents (code available).

**Strengths:**

The strengths of this paper can be summarized as,

- The authors open-source the code to facilitate readers’ reproduction (replication) of their results.

- This paper formalizes personalized function calling (tool, parameter, temporal), present TRACEBench built from real human histories to evaluate all three, and reports extensive experiments that benchmark LLMs and highlight challenges and future directions.

**Weaknesses:**

The main weaknesses / questions of this paper can be summarized as,

- Section 3.1 contains numerous imprecise and erroneous descriptions, *e.g.*,
     - They define the candidate set as $ \mathcal{T} = \{(d(t_1), ..., d(t_N))\} $ (docs), but then write $ t \in \mathcal{T} $. That’s inconsistent: $ \mathcal{T} $ should be tools, not their docs.
     - $ p $ should be a parameter assignment/argument tuple drawn from a tool-specific domain, not an untyped “set”.
     - The argmax domain ignores that the admissible parameters depend on the chosen tool.
     - If documentation is used (as it typically is), it should appear in the conditioning; otherwise the probability term is underspecified.
     - Equation (2) does not incorporate temporal information.

- Section 3.2 contains numerous imprecise and erroneous descriptions, *e.g.*,
     - “Functionally equivalent tools” is used but no equivalence relation is specified, so the candidate subset is ill-posed.
     - “We term this phenomenon as tool personalization” → “We term this phenomenon tool personalization.”
     - The text says preferences depend on non-functional attributes (price, speed, service) and user context/history, but none of these variables appear in the notation.
     - The relation $\succ_{(u,q)}$ has no stated
properties (e.g., totality, transitivity). If it is only a partial order, a unique maximizer need
not exist; tie-breaking is unspecified.

- The key issues in Section 3.3,
    - It’s unclear whether 𝐴 A is a set of parameter slots, a vector of values, or a tool-specific domain; parameters should be tool-dependent.
    - Using linear-algebraic “span” for text/structured history is ill-defined; information dependence should be probabilistic or via a measurable mapping.
    - No formal notion of time or decay; $\mathcal{H}_u$ lacks timestamps and recency weighting.
    - What happens if history is uninformative or conflicting is unspecified.
    - The probability/utility that selects parameters is missing.
    - Binary existence is too weak for $\alpha$. Moreover, real cases involve multiple parameter slots; notation should handle vectors and per-slot missingness.

- The key issues in Section 3.4,
    - The authors mention “short-term” but never define a window $k$ or decay; timestamps are absent.
    - The authors define $(c_i,o_i)$ but the rule uses only $\{p_{r-k},\dots,p_{r-1}\}$; outputs $o_j$ (often the key temporal signal) are ignored.
    - Writing $\alpha\in p_r$ and $\alpha=f(\{p_{r-k},\dots,p_{r-1}\})$ mixes a scalar with heterogeneous parameter vectors across tools, without a mapping between parameter spaces.
    - “Does not belong to long-term interests” lacks an operational criterion (no threshold/test to separate short-term from profile-based effects).
    - What happens if recent history is uninformative or conflicting is unspecified?

- In section 4.1, the authors say APIs are collected from real apps, but also that GPT-4o generates “initial designs and tool descriptions.” That makes the tools partly synthetic, undermining the claim of realism and threatening reproducibility/version drift.

- “8 scenarios” are asserted without selection criteria, diversity checks, or statistics (tools per domain, endpoints, arg types, auth modes). External validity is unclear (e.g., JD/Taobao only).

- “Manually check and optimize” lacks protocol, annotator count, QA criteria, and inter-rater agreement—hard to reproduce.

- Are calls live, mocked, or sandboxed? No mention of rate limits, failures, retries, caching, or version pinning; results won’t be replicable.

- The text mixes “APP level” with API endpoints; it’s unclear what the agent actually calls.

- No confirmation of ToS compliance, licensing for brand APIs, or mitigation of LLM prior knowledge leakage when GPT-4o drafts tools.

- The authors assert “authentic, anonymized interaction data from volunteers,” but give no cohort size, demographics, languages, collection duration, domains per user, or basic stats (sessions/user, tool-calls/session, length distributions). No recruitment method, inclusion/exclusion criteria, compensation, or bias analysis; external validity is unknown.

- “Temporal/logical consistency” lacks operational rules, metrics, annotator protocol, inter-rater agreement, or error rates; discarding “incomplete/corrupted” sessions may bias away from realistic edge cases.

- The authors don’t say how “true” tool choice/parameters/temporal dependencies are labeled or inferred from history.

- Mapping logs to a unified API schema can erase vendor-specific heterogeneity (args, defaults, failure codes) that tool personalization depends on; no audit of information loss.

Overall, introducing personalization is a promising idea; however, the paper reads rather rushed. Many parts feel awkward—especially the definitions of equations and concepts—while the experimental design is mediocre and the analysis lacks depth.

**Questions:**

The issues are mainly reflected in the weaknesses: the manuscript contains numerous unprofessional and non-rigorous descriptions, along with many incomplete sections; the authors are advised to substantially revise and improve the work before resubmission.

---

### Official Review · Reviewer_ar7G · 2025-11-01

**Soundness:** 3
**Presentation:** 2
**Contribution:** 3
**Rating:** 4
**Confidence:** 4

**Summary:**

This paper introduces TRACEBench, a benchmark designed to evaluate personalized function calling capabilities in Large Language Models. The authors identify three key dimensions of personalization: tool personalization (selecting between functionally similar tools based on user preferences), parameter personalization (inferring missing parameters from long-term user habits), and temporal context personalization (reasoning over short-term dynamic context to infer parameters). The benchmark is derived from user interaction collected from human volunteers, rather than synthetic LLM-generated data. The authors construct a dataset comprising 1,815 queries from 10 users across 8 scenarios. They evaluate various open-source and closed-source models, finding that current LLMs struggle significantly with inferring correct parameter values, particularly as query vagueness increases.

**Strengths:**

- The paper makes a valuable contribution by formally defining and distinguishing three types of personalization in function calling. The distinction between parameter personalization (long-term habits) and temporal context personalization (short-term dynamic context) is particularly insightful
- The use of real human interaction logs collected from volunteers makes the benchmark valuable. The manual review process for anchor behavior selection and query generation further adds credibility.

**Weaknesses:**

- The evaluation is restricted to a narrow range of models, with open-source models limited to the 7-8B parameter range. Given the importance of this benchmark, evaluation should include larger open-source models (13B, 70B variants) and additional API-based models to provide a more comprehensive assessment of the current state of personalized function calling capabilities.
- The low performance on format following of xLAM, ToolACE, and watt-tool models raises concerns about the evaluation setup. These models perform strongly on other benchmarks, such as BFCL, suggesting that they may not have been tested with their native chat templates.
- The paper would benefit from including sample data points with actual model outputs in an appendix. Showing concrete examples of queries at different vagueness levels, the corresponding ground truth function calls, and actual model generations would help readers better understand both the benchmark's challenges and the nature of model failures
- Figure 3 uses a radar chart to visualize the performance across different vagueness levels. It is difficult to interpret and compare across models. It would help if the authors could explore other ways of presenting the results, such as with a bar chart or in a table.
- The data collection methodology around real-user logs lacks crucial details, such as the recruitment criteria, geographic distribution, or demographic characteristics.. Without these details, concerns about selection bias and generalizability remain unaddressed.
- The paper will also benefit from providing some practical guidance on where function-calling models currently lack and how to improve them in the future. Exploration of various prompting strategies to improve performance will also be beneficial.

**Questions:**

See above

---

### Note · Authors · 2025-12-01

I have read and agree with the venue's withdrawal policy on behalf of myself and my co-authors.